# Assessing the impact of preventive mass vaccination campaigns on yellow fever outbreaks in Africa: A population-level self-controlled case series study

Kévin Jean [1,2,3]*, Hanaya Raad[1,2,4], Katy A. M. Gaythorpe[3], Arran Hamlet[3], Judith E. Mueller[2,4], Dan Hogan[5], Tewodaj Mengistu[5], Heather J. Whitaker[6,7], Tini Garske[3], Mounia N. Hocine[1]

1 Laboratoire MESuRS, Conservatoire national des Arts et Métiers, Paris, France, 2 Unité PACRI, Institut Pasteur, Conservatoire national des Arts et Métiers, Paris, France, 3 MRC Centre for Global Infectious Disease Analysis, Department of Infectious Disease Epidemiology, Imperial College London, London, United Kingdom, 4 EHESP French School of Public Health, Paris, France, 5 Gavi, the Vaccine Alliance, Geneva, Switzerland, 6 Statistics, Modelling and Economics Department, National Infection Service, Public Health England, Colindale, London, United Kingdom, 7 Department of Mathematics & Statistics, The Open University, Milton Keynes, United Kingdom

* kevin.jean@lecnam.net

**Data Availability Statement:** All data and codes used for the analysis are publicly available at https://github.com/kjean/YF_outbreak_PMVC.

## Abstract

### Background

The Eliminate Yellow fever Epidemics (EYE) strategy was launched in 2017 in response to the resurgence of yellow fever in Africa and the Americas. The strategy relies on several vaccination activities, including preventive mass vaccination campaigns (PMVCs). However, to what extent PMVCs are associated with a decreased risk of outbreak has not yet been quantified.

### Methods and findings

We used the self-controlled case series (SCCS) method to assess the association between the occurrence of yellow fever outbreaks and the implementation of PMVCs at the province level in the African endemic region. As all time-invariant confounders are implicitly controlled for in the SCCS method, this method is an alternative to classical cohort or case–control study designs when the risk of residual confounding is high, in particular confounding by indication. The locations and dates of outbreaks were identified from international epidemiological records, and information on PMVCs was provided by coordinators of vaccination activities and international funders. The study sample consisted of provinces that were both affected by an outbreak and targeted for a PMVC between 2005 and 2018. We compared the incidence of outbreaks before and after the implementation of a PMVC. The sensitivity of our estimates to a range of assumptions was explored, and the results of the SCCS method were compared to those obtained through a retrospective cohort study design. We further derived the number of yellow fever outbreaks that have been prevented by PMVCs. The study sample consisted of 33 provinces from 11 African countries. Among these, the

**Funding:** This work was carried out as part of the Vaccine Impact Modelling Consortium (www. vaccineimpact.org), but the views expressed are those of the authors and not necessarily those of the Consortium or its funders. The funders were given the opportunity to review this paper prior to publication, but the final decision on the content of the publication was taken by the authors. KJ, AH, KAMG, and TG acknowledge joint Centre funding from the UK Medical Research Council and Department for International Development (MR/ R015600/1) and report grant from The Bill & Melinda Gates Foundation (grant number OPP1157270). The funders had no role in study design, data collection and analysis, decision to publish, or preparation of the manuscript.

**Competing interests:** Tini Garske was the principal investigator of a grant from The Bill & Melinda Gates Foundation, and the principal investigator of a grant from GAVI, both funding the Vaccine Impact Modelling Consortium. Other authors have declared no competing interests. Author Tini Garske was unable to confirm their authorship contributions. On their behalf, the corresponding author has reported their contributions to the best of their knowledge.

**Abbreviations:** EYE, Eliminate Yellow fever Epidemics; IRR, incidence rate ratio; PMVC, preventive mass vaccination campaign; SCCS, self-controlled case series; WHO, World Health Organization.

first outbreak occurred during the pre-PMVC period in 26 (79%) provinces, and during the post-PMVC period in 7 (21%) provinces. At the province level, the post-PMVC period was associated with an 86% reduction (95% CI 66% to 94%, $p < 0.001$) in the risk of outbreak as compared to the pre-PMVC period. This negative association between exposure to PMVCs and outbreak was robustly observed across a range of sensitivity analyses, especially when using quantitative estimates of vaccination coverage as an alternative exposure measure, or when varying the observation period. In contrast, the results of the cohort-style analyses were highly sensitive to the choice of covariates included in the model. Based on the SCCS results, we estimated that PMVCs were associated with a 34% (95% CI 22% to 45%) reduction in the number of outbreaks in Africa from 2005 to 2018. A limitation of our study is the fact that it does not account for potential time-varying confounders, such as changing environmental drivers of yellow fever and possibly improved disease surveillance.

## Conclusions

In this study, we provide new empirical evidence of the high preventive impact of PMVCs on yellow fever outbreaks. This study illustrates that the SCCS method can be advantageously applied at the population level in order to evaluate a public health intervention.

## Author summary

### Why was this study done?

- Yellow fever is a mosquito-borne, vaccine-preventable disease that may cause large urban outbreaks, especially in tropical African regions.

- Since 2006, preventive mass vaccination campaigns (PMVCs) have been implemented in many African provinces. These are large-scale vaccination campaigns targeting all or most age groups in a specific area.

- The preventive impact PMVCs may have on the risk of yellow fever outbreak has not been quantified yet.

### What did the researchers do and find?

- We used the self-controlled case series (SCCS) method to assess the association between PMVCs and outbreak risk at the province level in 34 African countries between 2005 and 2018.

- We compared pre- and post-PMVC periods within each province individually, thus implicitly controlling for all possible confounders that do not vary in time, especially the fact that provinces indicated for PMVCs are generally those considered at highest baseline risk of yellow fever (confounding by indication).

- At the province level, we estimated that implementation of PMVCs was associated with an 86% reduction (95% CI 66% to 94%) in the risk of yellow fever outbreak.

- A complete cohort analysis provided less reliable results than the SCCS method, likely because of confounding by indication that was not entirely controlled for by adjusting for known drivers of yellow fever.

- We further estimated that all PMVCs conducted between 2006 and 2018 in Africa may have reduced the total number of yellow fever outbreaks by 34% (95% CI 22% to 45%).

### What do these findings mean?

- These results provide new empirical evidence of the high preventive impact of PMVCs on yellow fever outbreaks.

- These results may encourage rapid rescheduling of yellow fever PMVCs that have been postponed due to the COVID-19 pandemic.

- The SCCS design may be advantageously applied at the population level to assess the impact of public health interventions.

## Introduction

Recent years have seen the resurgence of yellow fever outbreaks in Africa and Latin America [1]. Regarding Africa specifically, 5 alerts have been issued for the first semester of 2020 alone (Uganda, South Sudan, Ethiopia, Togo, and Gabon) [2]. As a response to the large-scale Angola 2015–2016 outbreak, the World Health Organization (WHO) launched the Eliminate Yellow fever Epidemics (EYE) initiative in 2017 [3]. This strategy aims at preventing sporadic cases sparking urban outbreaks and potentially triggering international spread. It relies on various vaccination activities, including preventive mass vaccination campaigns (PMVCs) that target all or most age groups in a specific area. Evaluating the health impact of such campaigns is key for informing further PMVCs within or beyond the EYE strategy, to ensure population acceptance and adherence to vaccination campaigns, and to sustain domestic and international efforts for vaccination activities.

Previous attempts have been made to estimate the impact of vaccination activities, including PMVCs [4–6]. These attempts mostly relied on mathematical models to estimate PMVC impact in terms of deaths or cases prevented over the long term. However, few studies aimed at quantifying the effect of vaccination campaigns on the risk of outbreak. Regardless of the number of cases they may generate, outbreaks can possibly lead to healthcare, economic, and social destabilizations of entire regions. As an example, the West African 2013–2016 Ebola outbreak strained health systems. This caused excess deaths due to neglected malaria control [7,8].

When assessed at the population level, the association between vaccination activities and risk of outbreak can be approached within a classical epidemiological perspective. In the same way that individual participants are followed up in cohort studies, populations (e.g., populations living in well-defined geographical areas) can be followed over time while tracking both population-level exposure (implementation of vaccination activities) and events (outbreaks). In such observational studies, a risk of confounding arises when both exposure and event share a same cause. This risk is high when measuring the association between PMVCs and

yellow fever outbreaks because PMVCs usually target areas that are assessed as being at particularly high risk by public health officials, due to disease circulation in the past or based on expert view or risk assessment [9]. Such a risk of confounding by indication is usually overcome in the statistical analysis by conditioning on (generally by adjusting for) the shared common cause; in this case the baseline risk of yellow fever in the area. However, the environmental or demographic drivers of yellow fever are not fully understood [10,11], leading to a situation in which residual confounders may bias the measure of association.

The self-controlled case series (SCCS) method is a case-only epidemiological study design for which individuals are used as their own control [12]. As all known and unknown time-invariant confounders are implicitly controlled for, the method is a relevant alternative to classical cohort or case–control study designs when the risk of residual confounding is high. The SCCS method has successfully been applied at the individual level, comparing exposure versus non-exposure periods within individual cases [13]. However, to our knowledge, this method has never been used for population-level case series to evaluate the health effects of a public health intervention in specific regions, countries, or other predefined geographical clusters that may be considered as group-level cases.

Here, we illustrate the use of the SCCS method at the population level by assessing the association between the implementation of PMVCs and the occurrence of yellow fever outbreaks at the province level in the African endemic region between 2005 and 2018.

## Methods

### Study hypotheses

Considering the yellow fever vaccine's high level of efficacy [14], and given the fact that PMVCs target all or most age groups in targeted areas, we expected to detect a substantial preventive effect of PMVCs on the risk of outbreak. We also expected to detect this association in a cohort design, providing confounders in the association between exposure to PMVCs and outbreaks were adequately controlled for (no model misspecification). A SCCS model would avoid the risk of residual confounding, at least for time-independent variables, but would reduce statistical power as compared to a cohort design analysis [15].

### Data used

This study relied on datasets that were previously collected and regularly updated for a broader project aiming at estimating the burden of yellow fever and the impact of vaccine activities [4,6]. The analytic plan was defined before the start of the analysis. This study did not require ethical approval.

In 2005, the WHO Regional Office for Africa established a yellow fever surveillance database across 21 countries in West and Central Africa based on reports of suspected yellow fever cases [4]. The establishment of this database is likely to have influenced the standards of yellow fever surveillance in these countries. In order to reduce the possible effect of time-varying surveillance quality, the beginning of the study period was set at this date. We compiled locations and dates of yellow fever outbreaks reported in Africa between 1 January 2005 and 31 December 2018 from international epidemiological reports, namely the WHO Weekly Epidemiological Record (WER) and Disease Outbreak News (DONs) [16,17]. As per WHO recommendations, a single, confirmed yellow fever case is sufficient to justify outbreak investigation. Based on the results of the investigation and on the absence of other suspected cases, an alert can be classified as an isolated case; such cases were not considered for this study [18]. Locations were resolved at the first subnational administrative level, hereafter called province, and

data were recorded for each outbreak with the date of occurrence. Outbreak reports that could not be located at the province level were excluded.

We compiled data regarding PMVCs conducted as part of the Yellow Fever Initiative since 2006 [19], and additional campaigns conducted under the EYE strategy [1]. Starting dates and locations of PMVCs were collected, the resulting list of vaccination campaigns was compared with data from the WHO International Coordinating Group (ICG) on Vaccine Provision, and discrepancies were resolved. This virtually ensured completeness of information regarding mass vaccination activities. Note that information on the vaccine strain used for PMVCs was not widely available; however, the 2 vaccine strains recommended by WHO (17D and 17DD) do not differ in terms of immunogenicity [14,20]. PMVCs considered here involved full-dose vaccine only, as fractional-dose vaccination has been approved by WHO only as part of an emergency response to an outbreak if there is a shortage of full-dose yellow fever vaccine [21].

Estimates of population-level vaccine-induced protection against yellow fever were obtained from Hamlet et al. [22]. These estimates were obtained by compiling regularly updated vaccination data from different sources (routine infant vaccination, reactive campaigns, PMVCs) and inputting these into a demographic model.

**Main SCCS analysis.** For our main analysis, we used the SCCS method to estimate the incidence rate ratio (IRR) of yellow fever outbreak after versus before the implementation of a PMVC. We used the province as the unit of analysis, so that the main outcome represents the risk for a province to be affected by an outbreak. As the dependence between potential outbreak recurrences in the same province could not be excluded, we limited the analyses to the first outbreak occurrence per province for the main analysis [23]. We used a conditional Poisson model with logit link to model the occurrence of outbreaks [15].

Provinces included in the SCCS analysis were those both affected by an outbreak and targeted for a PMVC over a study period from 1 January 2005 to 31 December 2018. We defined the unexposed period as the pre-PMVC period. Previous research found that a single dose of yellow fever vaccine provides long-lasting immunity with high efficacy [14,20]. Therefore, and given the relatively short observation period of the study (14 years), we assumed the exposure period started at the date of the first PMVC and lasted until the end of the observation period. This assumption was made regardless of estimated achieved coverage or intra-province geographic extent of the campaigns. In other words, we assumed the campaigns achieved uniform high coverage in all age groups across provinces and that this high coverage was maintained up to the end of the study period. This assumption of persisting high coverage is further justified by the inclusion of yellow fever vaccine in the Enhanced Programme on Immunization of most of the countries in the study region [24]. Routine infant vaccination may not rapidly increase population-level immunity by itself, but is critical for maintaining high levels once achieved by the means of PMVCs.

**Alternative SCCS models and sensitivity analyses.** In order to allow for possible variation in the coverage achieved across PMVCs, we considered estimated population-level vaccine coverage as an alternative time-varying, quantitative exposure (considered as categories with 20% bandwidth: 0%–19%, 20%–39%, 40%–59%, 60%–79%, and 80%–100%).

We also used alternative SCCS models to assess the influence of several assumptions on our results (Table 1) [12]. We conducted an SCCS analysis considering all outbreaks, instead of the first one only, in order to evaluate the influence of the assumption of non-independent recurrence. Additionally, as it is possible that the occurrence of an outbreak could affect subsequent exposure, we conducted an SCCS analysis including a 3-year pre-exposure period.

As the precise dates of outbreaks and PMVC initiation were not always available, we assumed, where missing, that outbreaks started in the middle of the year and that exposure to

**Table 1. Analysis plan for the measure of the association between yellow fever vaccination activities and outbreak risk.**

| Model | Outcome | | Exposure | | | Covariates | | |
|---|---|---|---|---|---|---|---|---|
| | First outbreak only | Repeated outbreaks included | Binary exposure: Pre- versus post-PMVC | Inclusion of a 3-year pre-exposure period | Quantitative estimates of population-level vaccination coverage | None (self-controlled) | Covariates used in a previous statistical model | Covariates used in a previous mechanistic model |
| SCCS model 1 (main analysis) | X | | X | | | X | | |
| SCCS model 2 | | X | X | | | X | | |
| SCCS model 3 | X | | | X | | X | | |
| SCCS model 4 | X | | | | X | X | | |
| Cohort model 1 | X | | X | | | | X | |
| Cohort model 2 | X | | | | X | | X | |
| Cohort model 3 | X | | X | | | | | X |
| Cohort model 4 | X | | | | X | | | X |

PMVC, preventive mass vaccination campaign; SCCS, self-controlled case series.

PMVCs started at the end of the year. The influence of these assumptions was explored in sensitivity analysis.

Furthermore, to assess how the choice of the study period influenced our results, we also conducted a sensitivity analysis considering alternative start and end dates: (i) 1 January 2007 to 31 December 2018 and (ii) 1 January 2005 to 31 December 2014.

Lastly, to assess whether spatial autocorrelation could have affected our results, we conducted multiple resampling. In each resampling from the SCCS study sample, we only sampled 1 random province per country and reestimated the IRR of the association between exposure and the event. This approach implicitly accounts for spatial autocorrelation within, but not across, countries.

**Analysis using the cohort design.** We compared the results obtained using the SCCS method with those obtained using a classical cohort design. The study sample consisted of all provinces belonging to the 34 African countries at high or moderate risk for yellow fever [3]. We used univariate and multivariate Poisson regression models with robust variance, considering exposure alternatively as a binary (pre- versus post-PMVC) or continuous (vaccination coverage) time-dependent variable.

In a cohort design, the choice of covariates to include is critical to prevent bias due to residual confounding. However, there is currently no clear consensus about the demographic and environmental drivers of yellow fever. We thus considered 2 (partially overlapping) sets of covariates (S1 Text). Both sets of variables were documented to reproduce well the presence and absence of yellow fever records at the province level. The first set of covariates was previously used in a statistical model, whereas the second was used in a mechanistic model [4,10]. Statistical models aim to describe the patterns of association between species (including infectious agent species) and environmental variables, while mechanistic models aim at explicitly

representing biological processes in species' occurrence [25]. The association between each covariate and exposure status was explored using modified Poisson regression.

## Number of outbreaks averted and prevented fraction

For each province $i$, we estimated the expected number of outbreaks averted by PMVC, $A_i$, using the formula

$$A_i = \lambda_i(T - d_i)(1 - \text{IRR}) \tag{1}$$

where $T$ is the total time of observation, $d_i$ is the time at which the PMVC was implemented (if no PMVC in province $i$, then $d_i = T$), $\lambda_i$ is the rate of outbreak occurrence in a Poisson process in the absence of PMVC, and IRR is the incidence rate ratio after versus before PMVC implementation. With $N_i^{E-}$ being the number of outbreaks observed in province $i$ during the pre-PMVC period, an estimator of $\lambda_i$ is $\hat{\lambda}_i = N_i^{E-}/d_i$, which leads to

$$\hat{A} = \sum_i \frac{N_i^{E-}(T - d_i)}{d_i}\left(1 - \widehat{\text{IRR}}\right) \tag{2}$$

We obtained 95% confidence intervals for $A$ using bootstrap resampling (10,000 samples). For each sample, a value of IRR was randomly sampled based on the parameters estimated in the SCCS analysis.

Finally, based on $\hat{A}$ and $N$, the total number of outbreaks observed, we obtained the outbreak prevented fraction, PF, with

$$\text{PF} = 1 - \frac{N}{N + \hat{A}} \tag{3}$$

## Results

### Outbreak occurrence and PMVCs

A total of 96 outbreaks out of 97 records (99.0%) were geographically resolved. Among the 479 provinces in 34 countries within the African endemic or at-risk region for yellow fever, 81 provinces (16.9%) in 18 countries experienced at least 1 yellow fever outbreak between 2005 and 2018 (Fig 1A), including 12 provinces experiencing more than 1 outbreak. The Poisson probability distribution applied satisfactorily to the observed outbreak distribution (S1 Table). Overall, 124 (25.9%) provinces were targeted for at least 1 PMVC (Fig 1B). The SCCS study sample consisted of 33 (6.9%) provinces having experienced both outbreak and PMVC implementation over the study period. Temporal patterns in the estimated population-level vaccination coverage for this sample are displayed in S1 Fig. The median difference between the post- and the pre-PMVC estimate of vaccination coverage was 24.2 percentage points (interquartile range: 9.4–42.7) (S2 Fig).

### SCCS analysis

In the SCCS study sample, the first outbreak occurred during the pre-PMVC (unexposed) period in 26 (78.8%) provinces, and during the post-PMVC (exposed) period in 7 (21.2%) provinces (Fig 2). Under baseline assumptions, this corresponded to a significantly reduced IRR of 0.14 (95% CI 0.06–0.34) for the exposed versus unexposed periods. A similar protective association was observed when considering all outbreaks instead of only the first outbreak

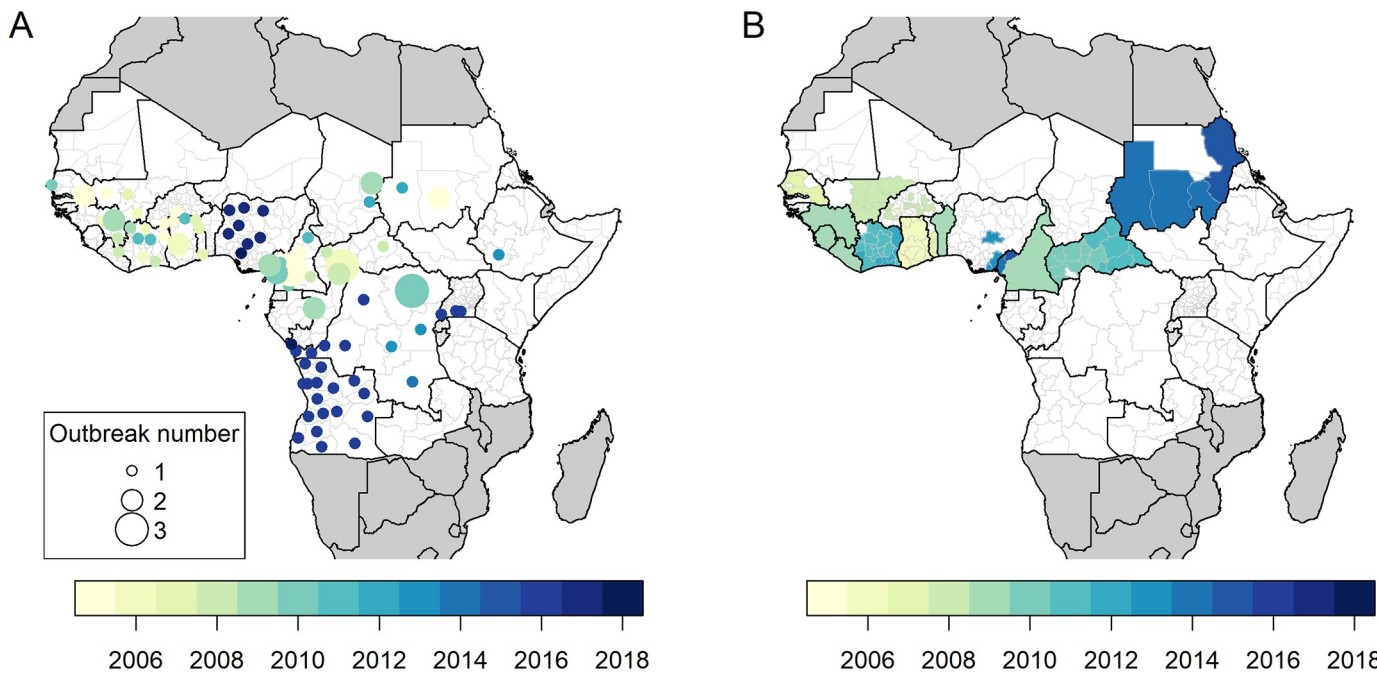

**Fig 1. Occurrence of yellow fever outbreaks and preventive mass vaccination campaigns at the province level over the 2005–2018 period.** (A) Yellow fever outbreaks. (B) Preventive mass vaccination campaigns. Maps were produced using GADM version 2.0.

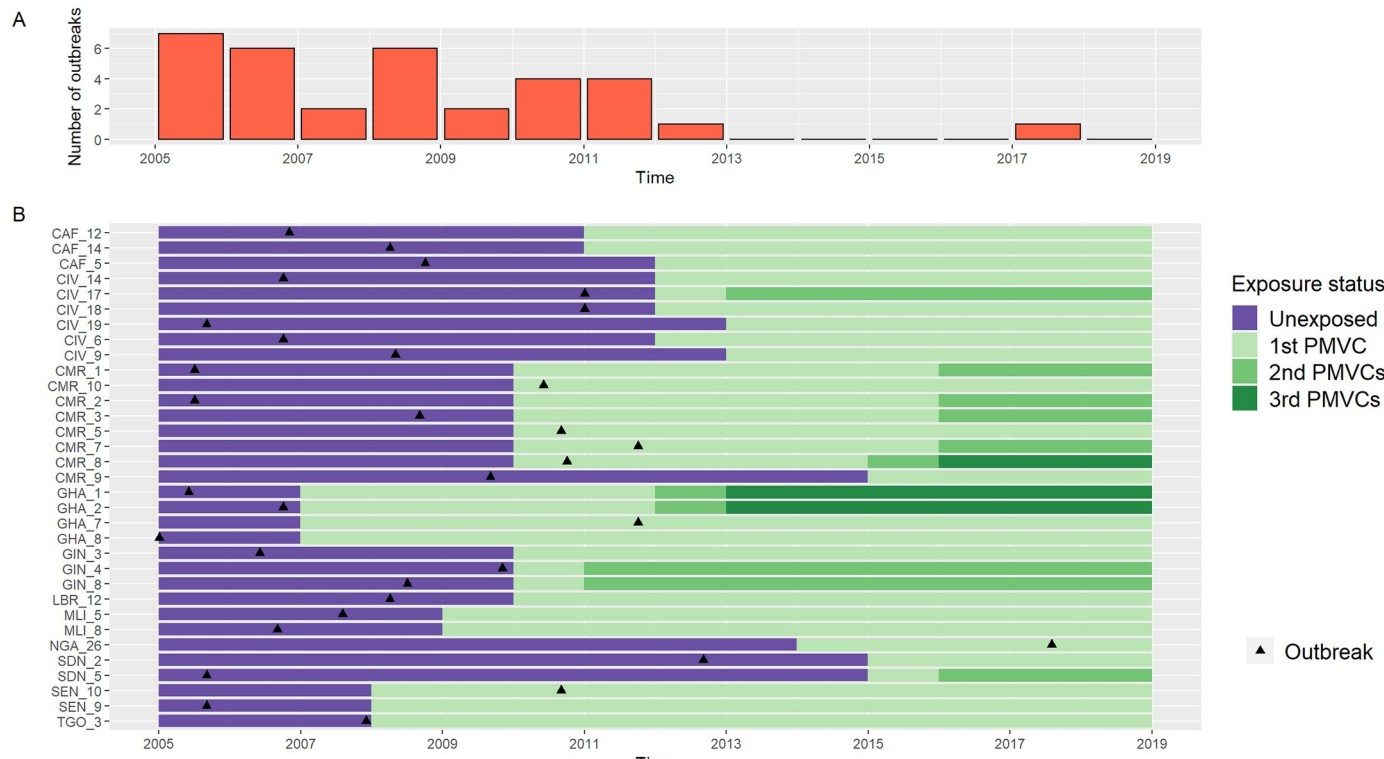

**Fig 2. Swimmer plot of the chronology of exposure to preventive mass vaccination campaigns (PMVCs) and yellow fever outbreaks among the 33 African provinces both affected by an outbreak and targeted for a PMVC (2005–2018).** (A) Time distribution of yellow fever outbreaks. (B) Swimmer plot. The 3-letter codes on the *y*-axis refer to International Organization for Standardization county codes (see S2 Table for complete province and country names).

**Table 2. Association between exposure to preventive mass vaccination campaigns and yellow fever outbreak in African provinces, 2005–2018.**

| Model | Exposure category | Number of events | IRR* | 95% confidence interval |
|---|---|---|---|---|
| SCCS model 1 (main analysis) | Unexposed (ref.) | 26 | 1.00 | — |
| | Exposed | 7 | 0.14 | 0.06–0.34 |
| SCCS model 2 (all outbreaks) | Unexposed (ref.) | 32 | 1.00 | — |
| | Exposed | 11 | 0.19 | 0.09–0.39 |
| SCCS model 3 | Unexposed (ref.) | 11 | 1.00 | Ref. |
| | Pre-exposed (3 years) | 15 | 0.99 | 0.42–2.30 |
| | Exposed | 7 | 0.14 | 0.05–0.40 |
| SCCS model 4 | Vc < 0.2 | 6 | 0.61 | 0.11–3.27 |
| | 0.2 ≤ Vc < 0.4 | 10 | 2.40 | 0.61–9.41 |
| | 0.4 ≤ Vc < 0.6 (ref.) | 8 | 1.00 | — |
| | 0.6 ≤ Vc < 0.8 | 5 | 0.29 | 0.06–1.41 |
| | 0.8 ≤ Vc ≤ 1 | 4 | 0.05 | 0.01–0.28 |
| Cohort model 1 (statistical model) | Unexposed (ref.) | 74 | 1.00 | — |
| | Exposed | 7 | 0.37 | 0.15–0.92 |
| Cohort model 2 (statistical model) | Vc < 0.2 | 19 | 0.18 | 0.07–0.50 |
| | 0.2 ≤ Vc < 0.4 | 40 | 0.86 | 0.43–1.73 |
| | 0.4 ≤ Vc < 0.6 (ref.) | 13 | 1.00 | — |
| | 0.6 ≤ Vc < 0.8 | 5 | 0.49 | 0.17–1.40 |
| | 0.8 ≤ Vc ≤ 1 | 4 | 0.11 | 0.03–0.36 |
| Cohort model 3 (mechanistic model) | Unexposed (ref.) | 74 | 1.00 | — |
| | Exposed | 7 | 0.65 | 0.26–1.65 |
| Cohort model 4 (mechanistic model) | Vc < 0.2 | 19 | 0.07 | 0.03–0.18 |
| | 0.2 ≤ Vc < 0.4 | 40 | 0.77 | 0.40–1.48 |
| | 0.4 ≤ Vc < 0.6 (ref.) | 13 | 1.00 | — |
| | 0.6 ≤ Vc < 0.8 | 5 | 0.47 | 0.16–1.40 |
| | 0.8 ≤ Vc ≤ 1 | 4 | 0.13 | 0.04–0.41 |

*For cohort models, IRR are adjusted on several demographic and environmental covariates, depending of the model (statistical or mechanistic), see S1 Text.

IRR, incidence rate ratio; SCCS, self-controlled case series; Vc, population-level vaccination coverage.

(IRR 0.19, 95% CI 0.09–0.39) or when including a 3-year pre-exposure period (IRR 0.14, 95% CI 0.05–0.40).

Considering estimates of population-level vaccine coverage as a categorical variable allowed us to observe a reduced risk of outbreak for higher levels of coverage (Table 2). Considering vaccine coverage as a continuous linear exposure led to a better fit of the model (likelihood ratio test: $p = 0.44$). Doing so, we estimated that a 10% increase in vaccine coverage was associated with a decrease in risk of outbreak of 41% (IRR 0.59, 95% CI 0.46–0.76).

## Sensitivity analysis

The negative association between exposure to PMVCs and outbreak remained significant across a range of assumptions regarding the imputation of the date (within the same year) of PMVC implementation and outbreak starting date (when missing), and across various observation periods (S3 and S4 Tables).

When resampling the SCCS study sample 100 times while allowing only 1 sampled province per country, and after excluding resampling yielding to random 0 in the corresponding contingency table ($n = 16$ with no outbreak occurring during exposed periods, thus leading to an

infinite confidence interval surrounding the association measure), we obtained an averaged IRR of 0.09 (95% CI 0.01–0.62).

### Cohort-style analysis

In a cohort design, over the 81 outbreaks (first outbreaks only) that occurred over the study period, 74 occurred during the unexposed period versus 7 occurring in the exposed period. Most of the environmental covariates we explored were associated with exposure to PMVCs (Table B in S1 Text). Exposure to PMVCs was associated with a significant reduced risk of outbreak (IRR 0.37, 95% CI 0.15–0.92) when adjusting for the covariates obtained from a statistical model. When adjusting for covariates obtained from a mechanistic model, exposure to PMVC was not significantly associated with the risk of outbreak (IRR 0.65, 95% CI 0.26–1.65) (detailed results in Table D in S1 Text). For both sets of covariates, we observed an inverse U-shaped association between the estimates of vaccination coverage and the risk of outbreak, with the risk decreasing for the lowest and highest values of vaccination coverage (Table 2).

### Number of outbreaks averted and prevented fraction

Based on the value of IRR estimated in the main analysis (IRR 0.14, 95% CI 0.06–0.34), we estimated that PMVCs implemented over the study period averted in median 50 (95% CI 28-80) outbreaks. When considering the 96 outbreaks in total that occurred over the study period, this corresponds to a prevented fraction of 34% (95% CI 22%-45%).

### Discussion

In this paper, using the SCCS method, we quantified the preventive effect of PMVCs on the risk of yellow fever outbreak at the province level, documenting a 86% (95% CI 66% to 94%) reduction in the risk of outbreak occurrence for provinces that were targeted by a PMVC. This result was robust over a range of assumptions. When using an estimate of population-level coverage as the exposure, we also observed a dose–response preventive effect on the risk of outbreak. Considering the scale of PMVC implementation during the study period, this corresponded to an estimated 22% to 45% of outbreaks averted by PMVCs in Africa between 2005 and 2018. In the cohort design analysis, the association between PMVCs and outbreak was sensitive to the choice of adjustment variables. Moreover, we observed a puzzling U-shaped association between vaccination coverage and the risk of outbreak in the cohort analysis. Overall, these results suggest a risk of residual confounding that the SCCS method, but not the cohort design, could overcome, at least for time-independent confounders. To our knowledge, this is the first time an SCCS analysis was conducted at the population level.

Considering evidence of yellow fever vaccine efficacy at the individual level, a preventive effect of PMVCs on outbreak risk was indeed expected. This is why we think that the cohort analysis results may be biased by residual confounding, whereas we consider the results obtained from the SCCS method to be more trustworthy. Indeed, the indication of provinces for PMVCs partly relies on a risk assessment [3]. For the results of the cohort design analysis to be valid, one needs to account for all possible confounders in the association between PMVCs and outbreak. This is particularly challenging as the environmental and demographic drivers of yellow fever circulation are not fully understood yet [9]. Another result suggesting residual confounding in the cohort design analysis is the U-shaped relationship between vaccination coverage and outbreak risk. The yellow fever vaccine has not been introduced in large regions of East Africa yet, as the risk of yellow fever is usually considered as low, though existing (e.g., Kenya). This can yield a spurious negative association between a low level of vaccination coverage and outbreak risk when confounders are not controlled for. In the SCCS

analysis, we did observe a linear relationship in the expected association between vaccination coverage and outbreak risk, providing further evidence of reduced residual confounding as compared to the cohort analysis. Analyzing cases only, instead of the corresponding complete cohort, translates into a loss of efficiency, but previous work has shown that this loss is small, especially when the fraction of the sample experiencing the exposure is high [26]. Moreover, this loss of efficiency has to be weighed against a better control of time-invariant confounders. Previous examples have illustrated that the SCCS design is likely to produce more trustworthy results than the corresponding cohort analysis, especially when a strong indication bias is likely [27,28].

Although the SCCS method was originally developed to be used at the individual level, we ensured that our analysis complied with all the method's requirements [12,15]. Exposure and outcomes were ascertained independently. The list of PMVCs was compiled based on information provided by international funders. Outbreak occurrence data were compiled from WHO sources, which themselves compile outbreak notification from countries as per the 2005 International Health Regulations. The observation period was chosen in order to maximize the chance that cases experienced the exposure period. Indeed, our observation period started shortly before the launch of the Yellow Fever Initiative. The Yellow Fever Initiative boosted the use of PMVCs, which had before that been very rare since the 1960s [4,19]. The choice of the long and unlimited exposure period was based on evidence regarding the long-lasting protection conferred by the yellow fever vaccine, and the SCCS method has been previously used successfully while considering long and unlimited risk periods [29]. The application of the SCCS design at the population level certainly deserves further methodological assessments to ensure its robustness to specific issues when transposing it to the population level, especially those related to overdispersed or autocorrelated events. We hope that the present work will stimulate further studies characterizing the advantages and drawbacks of SCCS as compared to more classically used population-level designs, for instance interrupted time series.

Under the assumption of causality, the IRR we estimate represents the average effect for a province being targeted by a PMVC, which corresponds to the average treatment effect in the counterfactual framework. This average effect likely masks large heterogeneity in the local effect of a PMVC. Indeed, PMVCs occur in populations with various baseline levels of immunity, and they may achieve various levels of post-intervention coverage. The dose–response relationship we observed in the association between vaccination coverage and outbreak risk brought additional evidence for a causal link between PMVCs and reduced outbreak risk. When looking at higher values of vaccination coverage, it is notable that several outbreaks ($n$ = 4) occurred at estimated levels of vaccination coverage > 80%, an empirical threshold that has been often suggested as protective against outbreaks [30]. While keeping in mind all the limitations such province-based estimates of vaccination coverage may have (outbreaks could occur in small pockets with low vaccination coverage even in provinces with high coverage), these occurrences of outbreaks at estimated vaccination coverage > 80% can be viewed as an argument to ensure high vaccination coverages homogeneously in at-risk areas, and to sustain them after PMVCs by ensuring routine infant vaccination.

Relying on our estimate of the preventive effect of PMVCs, the timing of implementation of these PMVCs, and the number outbreaks observed during the study period, we further estimated that PMVCs have averted 28 to 80 outbreaks in Africa between 2005 and 2018, corresponding to a prevented fraction lying between 22% and 45%. Garske et al. previously estimated that vaccination campaigns conducted up to 2013 averted between 22% and 31% of yellow fever cases and deaths in Africa [4], while Shearer et al. estimated that all vaccination activities (including routine infant vaccination) conducted up to 2016 have averted 33% to 39% of cases [5]. Our estimates were in a comparable range, although direct comparison with

these model-based estimates is not straightforward. Indeed, the latter are expressed as proportions of all yellow fever cases, including sylvatic cases that are not linked to outbreaks. Preventing outbreaks of epidemic-prone diseases is critical for ensuring global health security, yet there are few empirical studies that quantify the impact of public health interventions like immunization on the risk of outbreaks.

A main limitation of our study is that it does not account for possible time-varying confounders. Environmental changes affecting vector-borne diseases have been documented across tropical Africa over the study period, probably the main one being changing land use such as deforestation [31,32]. More frequent intrusions of humans into forest and jungles, together with increasing human mobility between endemic and non-endemic areas, have also been suggested to have affected yellow fever risk in the recent period [33]. Similarly, recent international emphasis on yellow fever may have led to better surveillance of the disease in the recent years. However, these various factors are likely to have increased the risk of outbreaks and the probability of outbreak detection in the recent period, which overlaps with the post-PMVC period in our study sample. This may have led to an underestimate of the association between PMVCs and yellow fever outbreaks. Lastly, historical vaccination activities that occurred up to the 1970s may potentially act as a time-varying confounder. Indeed, the contribution of older people (those potentially exposed to these historical campaigns) to the population-level immunity may decrease over time by population renewal. However, the corresponding bias is likely limited, considering the population structure skewed towards younger individuals in the region considered here. Moreover, such a bias is likely to have led to an underevaluation of the association between PMVCs and yellow fever outbreaks. Indeed, decreasing population-level immunity would have increased the risk of outbreak during the more recent period, which corresponds to the post-PMVC period.

Previous quantifications of the outstanding health impact of vaccination activities have mainly focused on cases or deaths prevented, and have relied on mathematical models whose structures and assumptions may be difficult to understand by a non-expert audience, whether that be decision-makers or targeted populations [34,35]. Here we further document vaccination impact using an empirical, maybe more intuitive approach, thus allowing for a triangulation of methods to further document the beneficial impact of yellow fever vaccine campaigns. This method relies on data that are quite easily accessible. Thus, our method could be applied to other diseases for which PMVCs are implemented, such as polio, meningitis, or cholera. Due to the COVID-19 pandemic, WHO recommended temporary suspension of preventive campaigns while risk was assessed, and effective measures for reducing COVID-19 circulation were established. In consequence, regarding yellow fever specifically, 4 countries postponed vaccination campaigns [36]. Our results provide additional evidence to encourage a rapid rescheduling of these vaccine campaigns in order to prevent further outbreaks of preventable disease.

## Supporting information

**S1 Fig. Temporal pattern in the estimate of population-level vaccination coverage in 33 African provinces having experienced both yellow fever outbreaks and the implementation of preventive mass vaccination campaigns over the 2005–2018 study period.** Each province is represented by a unique color.
(TIFF)

**S2 Fig. Distribution of the difference in population-level vaccination coverage between the post- and pre-PMVC periods.** PMVC, preventive mass vaccination campaign.
(TIFF)

**S1 STROBE Checklist. STROBE checklist for observational studies.**
(DOCX)

**S1 Table. Fit of the Poisson probability distribution to outbreak data.** The simulated counts were obtained from 10,000 random realizations of a Poisson process of rate λ = 96/479, based on the total number of outbreaks observed among the sample of 479 provinces over the study period.
(DOCX)

**S2 Table. Correspondence table of provinces and International Organization for Standardization country codes from Fig 2 in the main text and complete country and province names.**
(DOCX)

**S3 Table. Sensitivity of the self-controlled case series method results to the imputation of the missing dates of events (outbreaks) or exposure (PMVC).** IRR, incidence rate ratio; PMVC, preventive mass vaccination campaign.
(DOCX)

**S4 Table. Sensitivity of the self-controlled case-series method results to the choice of start and end dates of the study period.** IRR, incidence rate ratio.
(DOCX)

**S1 Text. Cohort models and adjustment.**
(DOCX)

## Acknowledgments

The authors would like to express their sincere gratitude to Paddy C. Farrington, the developer of the SCCS method, for his careful reading of and useful input to this work.

## Author Contributions

**Conceptualization:** Kévin Jean, Hanaya Raad, Tewodaj Mengistu, Tini Garske, Mounia N. Hocine.

**Data curation:** Kévin Jean, Katy A. M. Gaythorpe, Arran Hamlet, Tini Garske.

**Formal analysis:** Kévin Jean, Hanaya Raad.

**Funding acquisition:** Tini Garske.

**Investigation:** Kévin Jean, Hanaya Raad.

**Methodology:** Kévin Jean, Hanaya Raad, Mounia N. Hocine.

**Visualization:** Kévin Jean.

**Writing – original draft:** Kévin Jean.

**Writing – review & editing:** Hanaya Raad, Katy A. M. Gaythorpe, Arran Hamlet, Judith E. Mueller, Dan Hogan, Tewodaj Mengistu, Heather J. Whitaker, Tini Garske, Mounia N. Hocine.

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
