## [Editor Report · Decision Letter 0]

23 Jul 2020

Dear Dr Jean, 

Thank you for submitting your manuscript entitled "Assessing the impact of preventive mass vaccination campaigns on yellow fever outbreaks in Africa : a population-level self-controlled case-series study" for consideration by PLOS Medicine.

Your manuscript has now been evaluated by the PLOS Medicine editorial staff and I am writing to let you know that we would like to send your submission out for external peer review.

Kind regards,

Artur Arikainen,

Associate Editor

PLOS Medicine

---

## [Decision Letter · Decision Letter 1]

12 Oct 2020

Dear Dr. Jean,

Thank you very much for submitting your manuscript "Assessing the impact of preventive mass vaccination campaigns on yellow fever outbreaks in Africa : a population-level self-controlled case-series study" (PMEDICINE-D-20-03496R1) for consideration at PLOS Medicine. 

[LINK]

In light of these reviews, I am afraid that we will not be able to accept the manuscript for publication in the journal in its current form, but we would like to consider a revised version that addresses the reviewers' and editors' comments. Obviously we cannot make any decision about publication until we have seen the revised manuscript and your response, and we plan to seek re-review by one or more of the reviewers. 

We expect to receive your revised manuscript by Nov 02 2020 11:59PM. Please email us (plosmedicine@plos.org) if you have any questions or concerns.

We look forward to receiving your revised manuscript. 

Sincerely,

Emma Veitch, PhD

PLOS Medicine

On behalf of Artur Arikainen, PhD, Associate Editor, 

PLOS Medicine

plosmedicine.org

*At this stage, we ask that you include a short, non-technical Author Summary of your research to make findings accessible to a wide audience that includes both scientists and non-scientists. The Author Summary should immediately follow the Abstract in your revised manuscript. This text is subject to editorial change and should be distinct from the scientific abstract. Please see our author guidelines for more information: https://journals.plos.org/plosmedicine/s/revising-your-manuscript#loc-author-summary

*In the manuscript, please clarify if the analytical approach followed here corresponded to one laid out in a prospective protocol or analysis plan? Please state this (either way) early in the Methods section.

*There are a few places in the text where causal language is used that may not be merited given the possibility for the results to be driven in part by residual confounding and other biases - eg, one such example - Results, "SCCS analysis" section - "Doing so, we estimated that a 10%-increase in vaccine coverage decreased the risk of outbreak by 41% (IRR 0.59; 95% CI 0.46 – 0.76)." - could be better phrased as "10%-increase in vaccine coverage was associated with a 41% decrease in the risk...". There may be other places in the text where similar phrasing needs to be changed. 

Comments from the reviewers:

Reviewer #1: "Assessing the impact of preventive mass vaccination campaigns on yellow fever outbreaks in Africa : a population-level self-controlled case-series study" studies the association between yellow fever outbreaks and preventive mass vaccination campaigns (PMVCs), in selected African provinces from between 2005 and 2018. The primary analytic method used is the self-controlled case series (SCCS), in which each individual province is used as its own control. This allows all time-invariant confounding variables for each province to be automatically accounted for, and is a novel application towards public health intervention, according to the authors. Together with various secondary analyses (summarized in Table 1), it is estimated that PVMCs reduced yellow fever outbreaks by about 34% over the study period.

However, one trade-off of SCCS - other than time-varying confounders - is that the derived conclusions can vary significantly depending on the start and end times of the SCCS analysis (refer for example "Self-Controlled Case Series Methodology", Annual Review of Statistics and Its Application, Whitaker & Ghebremichael-Weldeselassie, 2019). The possibility of such start/end date bias, as well as some other pertinent considerations, might be addressed by the authors to lend further support to their findings:

1. As mentioned, a major concern is that the choice of start and end dates appears able to significantly affect the conclusions. In particular, from Figure 2, half of the pre-PVMC outbreaks occured in 2005 & 2006, with the other half occuring from 2007 to 2014. Although these provinces perform PVMC at different times on or after 2007, the 2007-and-after pre-PVMC incidence of yellow fever is roughly half of the pre-2007 period (13 cases in 116 province-year units, as compared to 13 cases in 66 previously), which may already be suggestive of the aforementioned time-varying confounders.

Moreover, consider if the start date was chosen to be 2007, or the end date as 2015 instead of 2019; it appears that the changes in IRR and T for Equation (1) would lead to possibly significant changes in the calculated prevented fraction, despite the underlying phenomena being the same.

The authors might discuss if there was any particular reason behind the choice of the 2005 start date (e.g. other than to be timed with the beginning of the first Yellow Fever initiative, from Line 330), or if an earlier start date might have been selected to provide better estimates for pre-PVMC incidence rate (from a longer sampling period). Relevant to this, secondary analyses ablating the analysis start and end dates might also be appropriate.

2. The completeness of outbreak/PMVC data might be further commented on; in Line 134, it is stated that "Outbreak reports that could not be located at the province level were excluded". How common is this occurence? From Line 135 on, it seems that PMVC data was based on two campaigns (the Yellow Fever initiative and EYE strategy). Would almost all PMVCs be reasonably expected to be covered in these sources? 

3. Also, YF mass vaccinations appear to have been common in many African regions from the 1930s through the 1960s (https://www.who.int/csr/disease/yellowfev/massvaccination/en/, also Line 331), and possibly thereafter. This appears to be a plausible important confounder.

4. The definition of "yellow fever outbreak" does not appear to be defined in the paper, and is only described as being compiled from the WER and DON. The specific criteria (e.g. Percentage of cases increase over some period? Percentage of population affected?) might be explicitly stated.

5. Also, it seems possible that some outbreaks may be significantly more serious/widespread than others; the provinces themselves appear to possibly be of widely different sizes/populations, such that near-concurrent outbreaks in multiple smaller provinces, might have been judged to be a single outbreak were they a single province. While this appears partially addressed by analysis on spatial autocorrelation, it would be good if the differing scales of outbreaks might be accounted for.

6. In Line 165, "categories with 20% bandwidth" might be explained further.

7. More details about the vaccinations might be provided if possible. Was the same vaccine (17D?) used for all provinces, were there any variations (e.g. fractional doses)? Would it be common for some provinces to have had ad-hoc vaccination/vaccination in infants?

8. The correspondence between an estimated prevented fraction of 22%-45%, and an aversion of 28-80 outbreaks, might be described in further detail.

9. In Table 2, the Unexposed (Ref.) of 32 for SCCS Model 2 might be further explained.

10. There are some minor grammatical issues (e.g. "started few time", Line 330; "28 to 80 outbreak", Line 351)

In summary, historical experience seems to broadly support the effectiveness of mass vaccination in preventing yellow fever outbreaks; however, the proposed quantification of this effect as presented in this manuscript through SCCS might stand to be further refined, in particular on potential sensitivity towards the choice of start and end dates.

Reviewer #2: ---

Review comments for manuscript ID: "PMEDICINE-D-20-03496", entitled "Assessing the impact of preventive mass vaccination campaigns on yellow fever outbreaks in Africa : a population-level self-controlled case-series study" of journal "PLOS Medicine".

Comments:

-Overall I think the cohort mechanistic model is more trust-worth, because it used all data (exposed and unexposed) and external factors (which is important).

- I think the SCCS model study focusing on the exposed provinces only is of limited value. Because they ignored the data of unexposed provinces. The potential influences of unexposed provinces on exposed provinces should be considered as well. Because the provinces (neighboring) should impact each other.

- How is the global climate changes on the YF outbreaks. I think unexposed provinces may be used as a ref. Another proxy, may be activity of other mosquito borne diseases. 

- Figure 2, please consider to add the total of outbreaks for each year as bar plot in the top of the swimmer plot.

-In all of these models, one would assume all countries or outbreaks are independent from each other? How confident is this assumption? Alternatively, I would think a mechanistic model for a large region (involve the hardest-hit countries) would show the effect of the control measure. Before and after control measure, the number of outbreaks changed over time for the whole region--The typos in Figure 1 legend should be corrected. 

--I suggest the authors to redraw some time series of outbreaks, for each country, for unexposed countries , for exposed countries. 

--I think some references were missing in the current study, see, for instance, https://www.nejm.org/doi/full/10.1056/nejmp1803433,

https://journals.plos.org/plosntds/article?rev=2&id=10.1371/journal.pntd.0006158,

https://www.sciencedirect.com/science/article/pii/S1386653214003692. 

Reviewer #4: This manuscript by Jean and colleagues uses a newly applied simple empirical method to estimate the effectiveness of preventative vaccine campaigns for Yellow Fever in Africa. The manuscript is well written, analysis clear and I do believe that the results do make a new contribution to the evidence base on effectiveness of the Yellow Fever vaccine. I do, however, have some concerns about the over simplicity of the method and its (previously) untested application to ecological data. Alas addressing many of these comments may be challenging given the limited data available. On balance, I do still think that this analysis is beneficial is appropriately caveated.

Major comments:

The authors state that "To our knowledge, this is the first time a SCCS analysis was conducted at the population level." This does raise some methodological robustness issues that would (in an ideal world) be explored in a more detailed statistical methods paper with a more appropriate dataset. Issues such as ecological fallacy, impact of over dispersed or autocorrelated events may have a significant impact on using the SCCS method at a population level, but with only 33 provinces under observation and only 7 post exposure events in this dataset I am sceptical whether such issues could be investigated. I appreciate that a full methodological deep dive with other datasets is probably beyond the scope of this manuscript, but perhaps the authors might want to downplay the emphasis on this being a proof of principle for the SCCS on population level data and instead emphasis the similarities between this and interrupted time series analyses that have a long history of application to population-level data.

Justifying the time-frame of protection. The analysis assumes PMVCs provide perpetual protection whereas we might assume protection to wane over time as new births and migration increase the susceptible proportion of the population. While I can see the advantages of using as much data as is available, I do think it would help if the authors could provide an a priori statement about the length of duration of protection from outbreaks that PMVCs are intended or expected to confer. 

Time-varying confounding- two un/under mentioned time-varying confounders that might be worth thinking about. First, naturally acquired immunity- especially as some of these PMVCs appear to be in response to recent outbreaks - ideally if more data were available accounting for seasonality and multi-annual cycles in outbreak occurrence would be helpful. Second, I assume (not a subject expert) that PMVCs also involve a number of preventative activities beyond vaccination e.g. vector control / environmental clear up, behavioural awareness, etc, etc that could also decrease the risk of an outbreak. Plausible time varying confounder in this case could also lead to an overestimation of PMVC effectiveness which might be worth mentioning.

Minor comments:

Suitability of representing outbreaks as a Poisson process- are they not over dispersed? (I appreciate difficult o answer empirically with this dataset)

Mechanistic model covariates not being associated in this analysis is interesting result- any thought as to why + implications for future mechanistic modelling work on this subject?

[LINK]

---

## [Decision Letter · Decision Letter 2]

10 Dec 2020

Dear Dr. Jean,

Thank you very much for re-submitting your manuscript "Assessing the impact of preventive mass vaccination campaigns on yellow fever outbreaks in Africa : a population-level self-controlled case-series study" (PMEDICINE-D-20-03496R2) for review by PLOS Medicine.

I have discussed the paper with my colleagues and the academic editor and it was also seen again by two reviewers. I am pleased to say that provided the remaining editorial and production issues are dealt with we are planning to accept the paper for publication in the journal.

[LINK]

We look forward to receiving the revised manuscript by Dec 17 2020 11:59PM.   

Sincerely,

Artur Arikainen

Associate Editor

PLOS Medicine

plosmedicine.org

Requests from Editors:

1. Please address the final reviewer comments below.

2. Lines 20-34, and 453-458: These can be deleted – these data (author contributions, competing interests, funding etc.) are taken from the online submission form.

3. Abstract:

a. Lines 42-43: Please reword to “However, by how much PMVCs are associated with a decreased risk of outbreak to occur has not yet been quantified.” (Your study design cannot prove causation.)

b. Line 47: Please list the main confounding factors accounted for.

c. Line 58: Please also include the total number of countries affected.

d. Lines 60 and 63: Please replace “reduced” with “associated with a reduction”, or similar . (Your study design cannot prove causation.)

e. Please quantify results with p values, as well as 95% CIs.

f. Please present data to 1 decimal place, to match the main Results section.

g. Please perhaps give the total number of infected individuals for the outbreaks, for reference, if known.

h. Line 62: Please list the main sensitivity analyses conducted.

i. Line 66: Please list another limitation, eg. possible unmeasured confounding factors.

j. Conclusion: Please start with “In this study, we observed that…”, or similar.

4. Author Summary: 

a. Please check the dates quoted, to match the main text.

b. Line 86: Please clarify “exposed and unexposed periods”, “same units of analysis”, and “known and unknown confounders”, for a lay reader.

c. Line 91: Please clarify “confounding by indication that was not entirely controlled for”, for a lay reader.

d. Line 98: “postponed”

5. Methods: If a prospective analysis plan (from your funding proposal, IRB or other ethics committee submission, study protocol, or other planning document written before analyzing the data) was used in designing the study, please include the relevant prospectively written document with your revised manuscript as a Supporting Information file to be published alongside your study, and cite it in the Methods section on line 160. A legend for this file should be included at the end of your manuscript. 

6. In the Methods, please state that this study did not require ethical approval.

7. Note from the Academic Editor: There is a minor note for lines 182-183 -- that's not exactly what WHO says about fractional dosing. Please do a very thorough review of the language and terms used, due to the precision of epidemiologic concepts used. For example, you state that the efficacy of the YF vaccines is known and the same for both 17D and 17DD -- this isn't quite right, it's not efficacy (there's no efficacy trial of YF vaccines).

8. Line 353: Please add “95% CI” in the brackets.

9. Figure 2B: If possible, please give the full province and country names on the vertical axis.

10. Please provide access details (DOI or URL) for reference 19.

11. Please mark preprint references as “[preprint]”.

12. When completing the STROBE checklist, please use section and paragraph numbers, rather than page numbers. 

Comments from Reviewers:

Reviewer #1: We thank the authors for addressing most of the issues raised in the previous review round, in particular the additional sensitivity analyses. It appears that historical data available for certain details (e.g. size of outbreak) is inherently not extremely reliable, with few prospects of additional clarification. However, acknowledging these caveats, the presented findings appear largely justified.

Nevertheless, the authors might consider explicitly including their treatment of Point 3 in the previous review round (on the [non-]confounding effect of past generations being mass-vaccinated) as an additional limitation, in the revised manuscript.

Reviewer #4: The authors have adequately addressed all my comments.

[LINK]

---

## [Editor Report · Decision Letter 3]

15 Dec 2020

Dear Dr Jean, 

On behalf of my colleagues and the Academic Editor, Rebecca F. Grais, I am pleased to inform you that we have agreed to publish your manuscript "Assessing the impact of preventive mass vaccination campaigns on yellow fever outbreaks in Africa : a population-level self-controlled case-series study" (PMEDICINE-D-20-03496R3) in PLOS Medicine.

PRESS

Sincerely, 

Artur A. Arikainen 

Associate Editor 

PLOS Medicine